# Vitiligo-like Lesions as a Predictor of Response to Immunotherapy in Non-Small Cell Lung Cancer: Comprehensive Review and Case Series from a University Center

João Queirós Coelho [1,*], Raquel Romão [1], Maria João Sousa [1], Sérgio Xavier Azevedo [1], Paula Fidalgo [1] and António Araújo [1,2,3,4]

1. Unidade Local de Saúde de Santo António, 4099-001 Porto, Portugal
2. Oncology Research Unit, 4050-346 Porto, Portugal
3. UMIB—Unit for Multidisciplinary Research in Biomedicine, 4050-346 Porto, Portugal
4. ICBAS—School of Medicine and Biomedical Sciences, University of Porto, 4050-313 Porto, Portugal
* Correspondence: joaoqueiroscoelho@chporto.min-saude.pt

**Abstract:** The reference to vitiligo-like lesions (VLLs) induced by immune checkpoint inhibitors (ICIs) as a valuable predictive marker of treatment success of immunotherapy with ICIs in melanoma has been mentioned in the literature. Its role in non-small cell lung cancer (NSCLC)-treated patients remains a poorly recognized phenomenon with uncertain significance regarding its predictive value. A retrospective, observational, single-center report was performed, with descriptive analysis of clinicopathological and treatment characteristics of patients with stage IV NSCLC who developed ICI-induced VLL between January 2018 and December 2022, contextualized in a comprehensive review of the literature and reported cases regarding this phenomenon. During the first 5 years' experience of ICI use in stage IV NSCLC treatment, three cases of ICI-induced VLLs were diagnosed. In line with the previous reports, two of the three presented cases exhibited treatment response and favorable prognosis. The recognition and understanding of the pathophysiological processes underlying ICI-induced VLLs may represent a promising opportunity to identify a predictive marker of tumor response to ICIs, with impact in treatment selection and patient management. It also may contribute to the recognition of new patterns of molecular expression that could lead to improvements in therapeutic development.

**Keywords:** vitiligo; adverse events; non-small cell lung cancer; immune checkpoint inhibitor; immunotherapy

## 1. Introduction

Immune checkpoint inhibitors (ICIs) targeting the programmed cell death-1/programmed death-ligand 1 (PD-1/PD-L1) pathway have significantly improved outcomes for patients with a variety of malignancies. In stage IV non-small cell lung cancer (NSCLC) without oncogenic drivers, ICIs have become the standard of care for first-line treatment (as monotherapy or in combination with chemotherapy) [1–3]. Their use in NSCLC has extended to stage III as consolidation therapy after definitive chemoradiotherapy [4], and recent evidence also suggests benefit in early stages like adjuvant and neoadjuvant treatment [5,6].

ICIs have a toxicity profile that differs from the side effects of common cytotoxic agents. Collectively recognized as immune-related adverse events (irAEs) and with the potential to affect any organ or system, these are related to the increased immune activation [7].

The association between irAEs and favorable tumor response to ICIs has been mentioned in the literature [8]. Specific attention is being drawn to the role of vitiligo-like

lesions (VLLs) as an independent predictive factor of treatment success in melanoma patients treated with ICIs [9–13].

Vitiligo is an acquired depigmenting disfiguring skin disease characterized by white, well-demarcated, uniform, asymmetric patches surrounded by normal skin. The occurrence of VLLs is a relatively common irAE during ICI treatment in melanoma patients, with reported incidence of up to 25% [14]. It has been suggested that ICI-induced VLLs during melanoma treatment are caused by T cell cross-recognition of antigens on both melanoma cells and normal melanocytes [10,15–18]. This idea comes into question as there are scattered reports describing ICI-induced VLLs in virtually all cancer types [19]. ICI-induced VLLs in NSCLC appear to be the most frequent among non-melanoma cancers [19], although they remain a poorly recognized phenomenon with uncertain significance regarding their predictive value [20–24].

Here, we present a comprehensive review of some of the most relevant published data regarding this subject, and the clinical characteristics and outcomes of all ICI-induced VLL stage IV NSCLC-treated patients during the first 5 years' experience of ICI use at a Portuguese university center.

## 2. Methods

A retrospective, observational, single-center report was performed, with descriptive analysis of clinicopathological and treatment characteristics of patients with stage IV NSCLC who developed ICI-induced VLLs between 1 January 2018 and 31 December 2022 in Unidade Local de Saúde de Santo António. Reports with inaccurate information regarding the assessment of treatment efficacy were excluded. Case reports of patients with previously diagnosed vitiligo were also not considered.

To perform a comprehensive review, data were collected from PubMed and Medline databases, in the form of case reports, case series, research or review articles regarding the development of ICI-induced VLLs during treatment of stage IV NSCLC. The evolution of reported patients and main conclusions of the data were summarized. The following keyword combinations were used to perform the search: vitiligo or adverse events and non-small cell lung cancer, immune checkpoint inhibitor or immunotherapy in the title, abstract or keywords. Articles published until 31 September 2023 and written in English were verified. Case reports of patients previously diagnosed with vitiligo or those in whom the clinical evolution of skin lesions or lung cancer was not completely clear were not included.

## 3. Results

### 3.1. Case Reports—Unidade Local de Saúde de Santo António's Experience

#### 3.1.1. Case 1

The patient was an 81-year-old male with an Eastern Cooperative Oncology Group (ECOG) performance status 1, a former smoker of 50 packs/year, with no personal history of autoimmune or cutaneous disease and a diagnosis of stage IV squamous cell (SCC) lung cancer with adrenal metastasis. PD-L1 expression was found in 1 to 5% of malignant cells (tumor proportion score, TPS). No testing on oncogenic drivers was performed due to a heavy smoking history. After multidisciplinary team discussion, first-line treatment was defined with chemotherapy (CT) with carboplatin and gemcitabine, resulting in Grade 3 (G3) fatigue after four cycles (common terminology criteria for adverse events version 5.0, CTCAE v5.0), leading to treatment discontinuation. The best response was partial response (response evaluation criteria in solid tumors version 1.1, RECIST v1.1). Disease progression was observed 3 months after first-line treatment discontinuation. Thus, the patient underwent second-line treatment with intravenous pembrolizumab (200 mg every 3 weeks during the first three cycles, and 400 mg every 6 weeks since the fourth cycle). Grade 2 (G2) hypothyroidism was identified after the fourth cycle, recognized as irAE after diagnostic work-up, leading to hormone supplementation and treatment continuation. The patient had symptomatic improvement with ICI treatment associated with sustained partial

response. After five pembrolizumab cycles, he presented macules and depigmented spots with well-defined limits on the hands and forearms, compatible with Grade 1 (G1) VLL (Figure 1). No other predisposing factors to de novo vitiligo were identified. No action towards VLL was taken. The patient completed 2 years of treatment with a preserved quality of life. Slight worsening of VLLs since their identification was noted (Figure 2). Progressive disease (PD) was documented 2 weeks after the conclusion of the preconized 2 years (35 cycles) of treatment with Pembrolizumab. Third-line treatment with docetaxel was proposed, with stable disease as the best response. CT was suspended after 8 months of treatment due to cumulative systemic toxicity (G3 fatigue) and best supportive care (BSC) was decided. The patient remained alive, with controlled symptoms and preserved quality of life at the last follow-up visit, 43 months after starting first-line therapy.

### 3.1.2. Case 2

The patient was a 63-year-old male with an ECOG performance status 1, a former smoker of 84 packs/year with no personal history of autoimmune or cutaneous disease and a diagnosis of stage IV lung adenocarcinoma with pulmonary and bone metastasis. We were unable to determine PD-L1 expression due to technical constraints. No oncogenic drivers were identified (analysis performed on tumor tissue sample through polymerase chain reaction testing EGFR and KRAS mutations). The patient underwent first-line treatment in a clinical trial regimen with intravenous carboplatin, paclitaxel and atezolizumab (four cycles) followed by maintenance therapy with intravenously atezolizumab (1200 mg every 3 weeks). The best response was partial response (RECIST v1.1). Approximately 33 months after the initiation of therapy (Cycle 47), depigmented spots dispersed by hands and forearms were noted, with initial progression followed by stability, compatible with G1 VLLs. No other predisposing factors to de novo vitiligo were identified. No action towards VLLs was taken. Among other irAE, transient G1 dermatitis was reported after Cycle 18, G1 asthenia was reported after Cycle 33, and transient G1 arthralgias, G1 diarrhea and G1 serum creatinine elevation were reported after Cycle 36. The patient presented a symptomatic benefit associated with sustained imagiological partial response. After 82 cycles and 54 months of treatment, oligoprogression with asymptomatic single pulmonary node was diagnosed, treated with stereotaxic radiotherapy, and maintained therapy with atezolizumab. After 94 cycles and 63 months of treatment, a new oligoprogression with an asymptomatic single cerebral lesion was identified and treated with a similar approach. Cerebral progression of the previously treated lesion was noted after 98 cycles and 69 months of treatment, conditioning performance status deterioration and suspension of systemic therapy. The patient was alive, with controlled symptoms and ECOG performance status 3 at the last follow-up, 70 months after starting first-line therapy.

### 3.1.3. Case 3

The patient was a 77-year-old male with an ECOG performance status 1, no history of tobacco use, no personal history of autoimmune or cutaneous disease and a diagnosis of stage IV lung adenocarcinoma with pulmonary metastasis. No PD-L1 expression in malignant cells was observed (0%, TPS). KRAS G12V mutation was identified using liquid biopsy-based next-generation sequencing (no tumor tissue sample available due to technical constraints). The patient underwent first-line treatment with intravenous carboplatin, pemetrexed and pembrolizumab (200 mg every 3 weeks). Evaluation of response showed progressive disease after four cycles (RECIST v1.1), with respiratory symptoms worsening. VLLs of hands and wrists were noted after two cycles of treatment. No other predisposing factors to de novo vitiligo were identified. No specific action towards VLL was taken. No other irAE were identified. The patient refused second-line therapy with a CT-based regimen and died in the sixth month of follow-up.

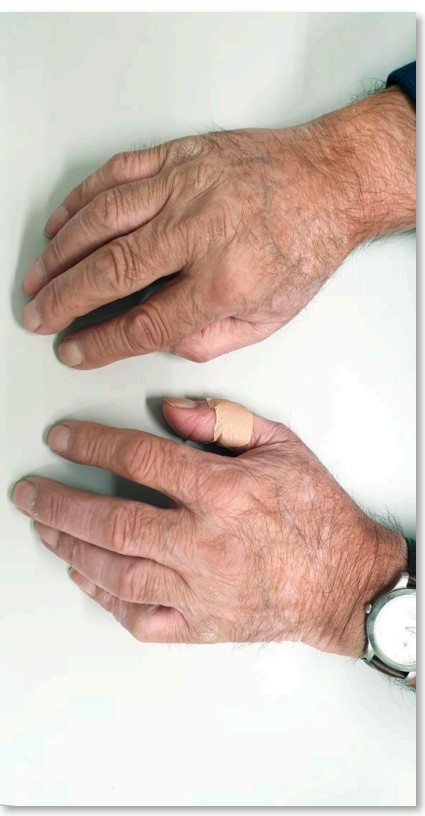

**Figure 1.** Vitiligo-like lesions in a patient receiving anti–programmed cell death-1 therapy (Case 1) after 4 months of treatment.

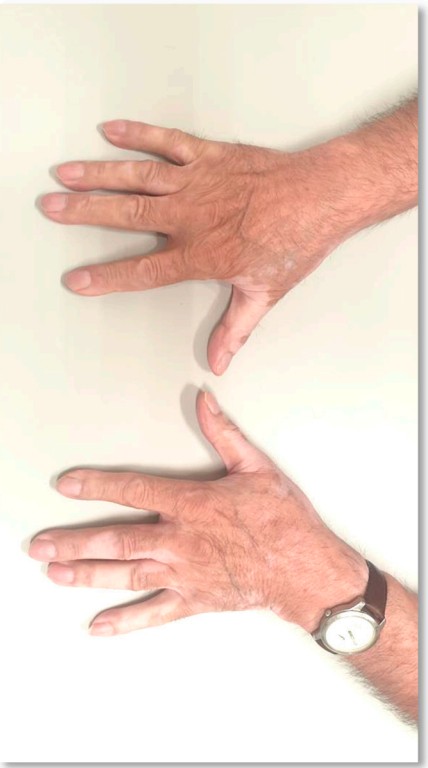

**Figure 2.** Vitiligo-like lesions in a patient receiving anti–programmed cell death-1 therapy (Case 1) after 13 months of treatment.

Table 1 summarizes the clinical features and outcomes of the reported cases.

**Table 1.** Summary of clinical features and outcomes of the all patients who developed ICI-induced VLLs during the first 5 years' experience of ICI use in stage IV NSCLC at a Portuguese university center.

| Case Report | TX Line | Histologic Subtype | ICI | PD-L1 (TPS) | Overall Response (RECIST v1.1) | Time to VLL Onset since ICI Initiation | Vitiligo (CTCAE v5.0) | Other irAE | Time to PD since ICI Initiation | Clinical State |
|---|---|---|---|---|---|---|---|---|---|---|
| 1 | 2nd | SCC | Pembrolizumab | 1–5% | PR | 5 months | G1 | G2 hypothyroidism | 24 months | Alive (43 months of f-up) |
| 2 | 1st | AC | Atezolizumab | UNK | PR | 33 months | G1 | G1 asthenia G1 arthralgias G1 diarrhea G1 SCE | 54 months | Alive (70 months of f-up) |
| 3 | 1st | AC | Pembrolizumab | 0% | PD | 2 months | G1 | None | 3 months | Death (6 months of f-up) |

AC, adenocarcinoma; CTCAE v5.0, common terminology criteria for adverse events version 5.0; F-up, follow-up; G1, Grade 1; G2, Grade 2; ICI, immune checkpoint inhibitor; irAE, immune-related adverse events; PD-L1, programmed death-ligand 1; PD, progressive disease; PR, partial response; RECIST v1.1, response evaluation criteria in solid tumors version 1.1; SCC, squamous cell carcinoma; SCE, serum creatinine elevation; VLL, vitiligo-like lesions; TPS, tumor proportion score; TX, treatment, UNK, unknown.

### 3.2. Previously Reported Cases

After applying the defined criteria mentioned in the Section 2, five published clinical cases of ICI-induced VLLs during treatment of stage IV NSCLC were identified [20–24]. Table 2 summarizes the relevant clinical features and respective outcomes.

**Table 2.** Summary of clinical features and outcomes of previously reported patients who developed ICI-induced VLLs during treatment of stage IV NSCLC.

| Case Reviewed | TX Line | Histologic Subtype | ICI | PD-L1 (TPS) | Overall Response (RECIST v1.1) | Time to VLL Onset since ICI Initiation | Vitiligo (CTCAE v5.0) | Other irAE | Time to PD since ICI Initiation | Clinical State |
|---|---|---|---|---|---|---|---|---|---|---|
| A | 4th | SCC | Nivolumab | UNK | PR/CR | 15 months | UNK | None | Still responding | Alive (34 months of f-up) |
| B | 1st | SCC | Pembrolizumab | UNK | PR/CR/SD | 2 months | UNK | G2 dermatitis | 10 months | Death (13 months of f-up) |
| C | 2nd | AC | Pembrolizumab | 70% | SD | 5 months | UNK | G2 hypothyroidism | 14 months | Alive (25 months of f-up) |
| D | 1st | AC | Pembrolizumab | 100% | CR | 3 months | UNK | None | Still responding | Alive (18 months of f-up) |
| E | 2nd | UNK | Nivolumab | 1–24% | PR | 20 months | UNK | G1 pneumonitis G1 arthralgias G1 hypothyroidism | Still responding | Alive (22 months of f-up) |

AC, adenocarcinoma; CR, complete response; CTCAE v5.0, common terminology criteria for adverse events version 5.0; F-up, follow-up; G1, Grade 1; G2, Grade 2; ICI, immune checkpoint inhibitor; irAE, immune-related adverse events; PD-L1, programmed death-ligand 1; PD, progressive disease; PR, partial response; RECIST v1.1, response evaluation criteria in solid tumors version 1.1; SCC, squamous cell carcinoma; SD, stable disease; VLL, vitiligo-like lesions; TPS, tumor proportion score; TX, treatment, UNK, unknown.

## 4. Discussion

### 4.1. Vitiligo and Lung Cancer Risk

In two analyses of Italian and South Korean patients, the presence of pre-existing vitiligo was linked to a reduced occurrence of melanoma, non-melanoma skin cancer and other solid malignancies [25–27]. Specifically, regarding the risk of development of lung cancer, the Asiatic study found a remarkably decreased risk of lung cancer in patients with vitiligo, with a hazard ratio (HR) of 0.75 (95% CI 0.59–076, $p < 0.001$). This effect was even more significant for male patients (HR 0.67, 95% CI 0.56–0.80) [26]. These data suggest that vitiligo might represent a biomarker of a systemically enhanced immune activity that would explain the reduced incidence of cancer in this population [25,26]. Similar conclusions have been reported by other studies, with data suggesting higher preponderance of this protective factor for the lung SCC histology subtype [28]. Authors highlighted the fact that, contrary to other common autoimmune diseases, vitiligo patients are submitted mainly to topical or local therapies, and not usually exposed to systemic immunomodulatory drugs such as cyclosporine, methotrexate, and tumor necrosis factor (TNF) inhibitors (commonly used to treat other autoimmune diseases). This could help explain the discrepancy with most previous studies regarding the increased risk of malignancies in patients with autoimmune diseases [25–27].

### 4.2. ICI-Induced VLLs

Cutaneous irAEs are the most frequently reported toxic effects of ICIs, occurring in 20% to 40% of all treated patients [29]. Vitiligo is a rare disease with a complex pathogenesis in which the loss of epidermal melanocytes occurs because of autoimmune activity [30]. Studies have identified the dysfunction of regulatory T cells ($T_{reg}$) and hyperactivation of $CD_8^+$ cytotoxic T cells as the main mechanism for melanocytes destruction [31,32]. It has been mentioned as one of the more frequent cutaneous toxicities associated with ICIs, after psoriasis, pruritus, macular rash, and eczematous-type reactions [13,19,29,33].

ICI-induced VLLs usually present with a distinct profile compared to classical vitiligo. They are typically characterized by a bilateral and symmetrical distribution pattern, with mostly smaller "freckle-like" macules, with a preferential distribution in ultraviolet-exposed areas (such as the face, hair, and hands), and with a median onset time of approximately 5 months. Melanoma patients with previous targeted therapy (with BRAF and MEK inhibitors) experience a significantly longer time lapse between ICI initiation and VLL onset compared to non-pretreated patients (12.5 vs. 6.2 months) [34,35]. Cytotoxic T-lymphocyte antigen-4 (CTLA4) inhibitor-induced VLLs were less frequent than they were with anti-PD-(L)1 treatment [19], although the combination of anti-PD-(L)1 and anti-CTLA4 usually results in a higher incidence of VLLs [9,19,36]. Reported cases of vitiligo-like depigmentation after ICI treatment include the following drugs: camrelizumab, QL1706 injection, sintilimab, tislelizumab, pembrolizumab, nivolumab, and ipilimumab [37].

The activation of immune checkpoints responsible for upregulation of diverse proinflammatory pathways is probably the trigger for the development of VLLs, as the PD-(L)1 pathway likely mediates peripheral tolerance of melanosome proteins [24].

Many reports have examined the prognostic impact of irAE regarding cancer outcomes, and even its predictive value in defining the probability of response to ICI. Most published data suggest that patients who developed irAE exhibited a reduced risk of death or progression among different primary tumors, namely malignant neoplasms of digestive organs, bronchus or lung, melanoma of skin, and urinary tract [38,39]. An exceptional significant association between irAE and ICI efficacy in patients with cancer has been attributed to dermatological events (vs. other irAEs), and to anti-PD-(L)1-treated patients (vs. CTLA4 inhibitors) [40]. Specifically, regarding cutaneous irAE, a retrospective cohort study with 15.000 patients suggested that the development of cutaneous irAE is strongly associated with response to ICI therapy and patient survival [29]. Although not reaching statistical significance, in this same work, ICI-induced VLLs were linked to a protective trend against mortality (HR 0.534, 95% CI 0.254–1.123) [29].

In melanoma patients treated with ICI, VLLs have been consistently referred to as an indirect predictive biomarker of favorable response to ICIs [10,14,35,41]. It is hypothesized that ICI-induced VLLs during melanoma treatment are caused by T cell cross-recognition of antigens on both melanoma cells and normal melanocytes (such as MART-1/MelanA, gp100, and tyrosinase-related Proteins 1 and 2) [10,15–18]. In support of this data, it was demonstrated that immunizing mice with human tyrosinase-related Protein 2 or utilizing the vaccinia virus to deliver their own tyrosinase-related Protein 1 prompts the development of vitiligo and imparts a shield of defense against melanoma. Furthermore, these models exhibit the presence of tissue-resident memory T cells, which provide localized protection against melanoma progression and endure within the areas where vitiligo manifests [25,42,43].

The reported data comparing histopathologic and immune profiles of classical vitiligo and ICI-induced VLLs revealed that, although both present partial or complete loss of melanocytes, infiltrates of T cells (usually absent in classical vitiligo) were found in ICI-induced VLLs [34,44], as well as higher proportion of produced interferon-$\gamma$ (IFN$\gamma$) and TNF$\alpha$, suggesting differences in $CD_8^+$ T cell modulation [45]. In support of that, inhibition of IFN$\gamma$ has been proved to reduce the accumulation of $CD_8^+$ T cells and prevent the loss of pigmentation [46]. Additionally, patients with ICI-induced VLLs presented with higher circulating levels of the CXCR3 ligand and CXCL10, potentially suggesting distinct mechanisms of cytotoxic cell infiltration and damage [47,48]. These features may play a significant role in enhancing anti-tumor immune response, and ultimately lead to cancer cell death [45,46]. A recent work characterized the behavior of immune cells during vitiligo onset in melanoma patients and revealed an association between a blood reduction in CD8-mucosal associated invariant T, T helper (Th) 17, natural killer (NK), CD56 bright, and T regulatory (T-reg) cells and vitiligo onset. Also, and consistently with the observed blood reduction in Th17 cells in melanoma patients developing ICI-induced VLLs, there was a high number of IL-17A-expressing cells in the vitiligo skin biopsy, suggesting a possible migration of Th17 cells from the blood into the autoimmune lesion. There were different T cell receptor sequences between vitiligo and primary melanoma lesions in most cases, although shared T cell receptor sequences were identified between vitiligo and metastatic tissues of the same patient. These data indicate that T cell response against normal melanocytes, which is involved in vitiligo onset, is not typically mediated by reactivation of specific T cell clones infiltrating primary melanoma but may be elicited by T cell clones targeting metastatic tissues. Altogether, these data indicate that anti-PD-1 therapy induces a de novo immune response, stimulated by the presence of metastatic cells, and composed of different T cell subtypes, which may trigger the development of vitiligo and response against a metastatic tumor [49].

Other distinct immune signatures were found among patients with VLL in comparison to ICI-treated patients without VLLs (irrespective of other irAEs), with selective upregulation of several proinflammatory receptors and kinases in the first group. A recent prospective observational single-center study highlighted a great heterogeneity and distinct proteomic profile among patients who developed VLLs during melanoma treatment with ICI. Comparison analysis between responders and non-responders' groups showed several increased proteins (such as EDAR and PLXNA4) and downregulation of others (such as LAG3) in the responders' group. Notably, integrin alpha 11 (ITGA11), a transmembrane protein that mediates cell adhesion to the extracellular matrix, known for its proinflammatory role, was identified as the most upregulated protein in patients with ICI-induced VLLs (compared to melanoma patients treated with ICI without VLLs) [34]. This finding gained preponderance as ITGA11 was also classified, in the same study, as the protein most closely associated with an improved overall survival in melanoma patients treated with ICI [34]. Based on these data, authors have hypothesized that a specific constellation of cytokines may induce or indicate VLLs and influence treatment outcomes [34].

Figure 3 summarizes some of the main mechanisms thought to be involved in the development of ICI-induced VLLs.

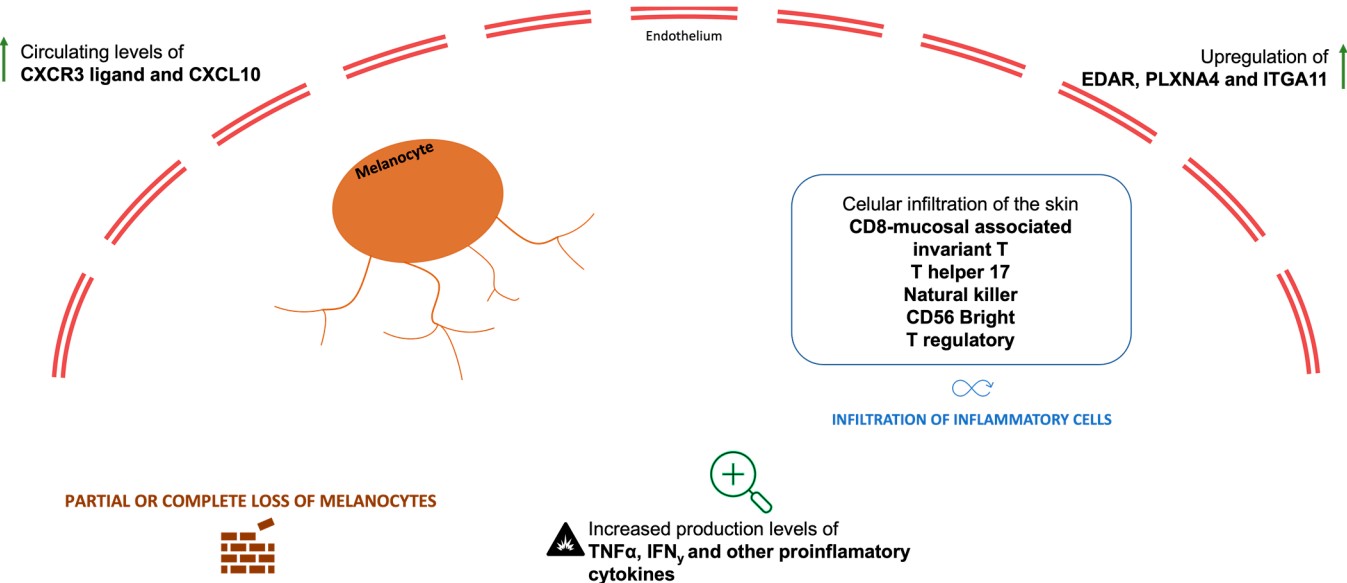

**Figure 3.** Most important mechanisms involved in the development of ICI-induced VLLs.

### 4.3. ICI-Induced VLLs and NSCLC

How the connotation between VLLs and response to ICI can be extrapolated from melanoma to be mirrored in NSCLC remains to be clarified. Some authors advocate the possibility of common antigens shared between healthy melanocytes, malignant melanoma, and NSCLC cells, which are targeted by activated T cells [20,21].

Regarding cutaneous irAEs in NSCLC, Hasan-Ali et al. specifically postulated patterns of lymphocytic skin infiltration that would differ depending on the histological tumor subtype. As lung SCC and keratinocytes both produce similar proteins including several cytokeratins, the histology of skin rashes of AC patients showed inflammation with lymphocytes located predominantly below the basal cell membrane within the dermis, which contains glandular structures that may share antigens with lung AC [50]. This may be particularly significant since, as previously referred to, vitiligo's apparent effect on reducing the probability of lung cancer seemed more relevant for SCC [28].

There are reports on ICI-induced VLLs on gastric adenocarcinoma, clear cell renal cell carcinoma and esophageal squamous cell carcinoma, which also appear to relate with prolonged progression-free survival, although this is based on a very limited number of cases [37]. Different authors have postulated that the development of this phenomenon could hold the same positive prognostic value in both melanoma and non-melanoma cancers [20,21,51].

### 4.4. Reflection on Reported Experience

The first five years' experience of ICI use in stage IV NSCLC at our center resulted in three cases of ICI-induced VLLs. Two of the three presented cases had a favorable response to ICI treatment, as the reported duration of response is superior to the median duration of the original clinical trials [52,53]. Regarding the reviewed cases, in all five, durable disease control was achieved, with the reported duration of response being superior (Cases A, C, D and E) or equal (Case B) to the median results of a respective clinical trial [20–24,52,53]. These results, despite being limited, appear to suggest a favorable association between ICI-induced VLLs and response to ICI also in NSCLC [20–24]. A small retrospective study also observed a positive correlation between skin irAEs (pruritus/rash) and tumor response in patients with nivolumab-treated NSCLC [50].

Case report 3 appears to display no response to treatment. The reason behind this outcome remains unknown. This patient exhibited KRAS G12V mutation. Although the standard first-line approach in patients with advanced, KRAS mutation-positive NSCLC is

based on ICI, with or without platinum-based CT [52,53], a greater concern rises regarding the effectiveness of ICI alone in this set of patients. Despite conflicting data, a recent meta-analysis suggests that KRAS mutations may represent a negative prognostic factor for survival and response outcomes to several different treatments (including ICI) in patients with advanced/metastatic NSCLC [54]. Retrospective data suggest that specific concurrent genomic alterations may predict ICI efficacy. Whereas co-occurring TP53 mutations are associated with inflammatory tumor microenvironment and higher levels of PD-L1 expression, STK11/LKB1 and KEAP1 mutations correlate with immunosuppressive tumor microenvironment and shorter survival [55,56]. Effectively, a recent retrospective series of KRAS-positive, advanced/metastatic NSCLC suggests an overall bad prognosis for this subgroup of patients despite the introduction of ICI in the therapeutic landscape [54]. An alternative explanation also relies on the possibility of another unidentified driver mutation predictive of no response to ICI, possibly unidentified by liquid biopsy considering its limitation in sensitivity compared to tumor samples [57]. This hypothesis is particularly relevant since the patient was a non-smoker, which is a recognized factor of increased probability to present oncogenic drivers [58].

Alongside with the data limitations (regarding their observational, retrospective character and the global number of patients), it is essential to maintain caution at the moment of reflecting on these outcomes and theories, as this association may be related to the lead-time bias: patients with disease progression are either switched to other treatment modalities or have a fatal outcome, while those who respond to ICI continue with more treatment cycles and thus have more time to develop and obtain reported cutaneous irAEs [23,33].

*4.5. Possible Clinical Practice Implications*

In the rapidly evolving landscape of cancer treatment, ICIs have emerged as a transformative approach, particularly in NSCLC [1,52,53,59–64]. Nevertheless, the success of ICI treatment is closely intertwined with the identification of predictive biomarkers [65,66]. These biomarkers play a pivotal role in stratifying patients, guiding treatment decisions, and optimizing therapeutic outcomes.

A positive association between the onset of VLLs and response to ICI therapy could act as an indirect tool to help clinicians predict treatment success and, with that, contribute to decision making in complex cases. More importantly, the understanding of the pathophysiological mechanisms associated with ICI-induced VLLs has the potential to elucidate new molecular predictive biomarkers for response to ICI treatment and novel strategies of modulating the inflammatory pathways to maximize immunotherapy success [21].

*4.6. Predictive Biomarkers of ICI Response in NSCLC*

Among the main identified predictive markers, PD-L1 expression, tumor mutation burden (TMB), tumor-infiltrating lymphocytes (TILs), neutrophil/lymphocyte ratio (NLR), and specific genetic mutations emerged as established or potential tools to tailor ICI-based strategies for NSCLC patients [65,66]. The prognostic value of these biomarkers is summarized in Table 3.

Despite its inaccurate and insufficient value, PD-L1 expression is still consensually referred to as the most current clinically impactful biomarker predictive of ICI response in NSCLC [65,66]. Trials like IMpower010, PACIFIC, KEYNOTE-024, KEYNOTE-042, IMpower110, CheckMate-026 and EMPOWER-Lung 1 are some of the most relevant studies that allowed understanding of that, for different defined cut-offs, with different drugs in different settings, higher values of PD-L1 expression tend to translate into greater benefit from ICI use or, eventually, differentiate from those without any clinical outcome advantage from ICI use [1,52,53,59–64]. Its limited role is, however, highlighted by the fact that responses are frequently seen in tumors without expression of PD-L1 [1,59]. Blood-based PD-L1 analysis (on soluble PD-L1, exosomal PD-L1 and PD-L1 in circulating tumor cells) also showed potential to predict treatment efficacy in NSCLC both before and after treatment with ICI. Although validation in clinical trial results remains needed, authors of

a meta-analysis suggested that pre-treatment soluble PD-L1 and exosomal PD-L1 act as unfavorable prognostic factors to patients with NSCLC undergoing ICI therapy, and that the dynamic upregulation of post-treatment exosomal PD-L1 levels indicates a favorable prognosis [67].

Research on TILs highlighted their crucial role in the immune system's anticancer activity and in the resistance mechanisms of certain tumors to ICI. Hummelink et al. established tumor-reactive tumor-infiltrating T lymphocytes as a predictive biomarker in advanced-stage NSCLC treated with PD-(L)1 blockade. Data showed that particularly low levels of tumor-reactive TILs accurately identify the patient's group with no clinical benefit from ICI. Furthermore, high tumor-reactive TILs infiltration was observed in most of the patients with durable responses [68].

The NLR reflects a state of systemic inflammatory response. Elevated NLR is also emerging as a promising and cost-effective predictive biomarker of response to ICI. In NSCLC, its role remains mainly as a prognostic factor in early-stage NSCLC treated with ICI. The Lung Immune Prognostic Index, based on the derived NLR and lactate dehydrogenase levels, was suggested as a predictive tool of resistance to ICI in advanced NSCLC [69]. Mezquita et al. also demonstrated that NLR was dynamic in 22% of patients and its early dynamic evolution significantly impacted ICI outcomes. A positive outcome was noted if NLR changed to low and a negative outcome if it changed to high [69].

The TMB is probably the most promising predictive biomarker for the efficacy of ICI treatment in NSCLC. Studies like CheckMate 227 suggested predictive value of high TMB ($\geq$10 mutations per megabase) to ICI response in NSCLC patients [3,70]. KEYNOTE-042 also demonstrated that tissue TMB can predict the efficacy of pembrolizumab monotherapy, with a cut-off of $\geq$175 mutations/exome [71]. A recent work calculated the molecular tumor burden index (mTBI) based on circulating tumor DNA and found that on-treatment mTBI dynamics were a predictor of long-term benefit from ICI. Decreasing levels of mTBI in ctDNA could be detected before the appearance of clinical or radiographic tumor shrinkage, which can help differentiate patients who will ultimately derive benefit from immunotherapy from those with stable disease or pseudo-progression in early evaluation [72].

Besides the already mentioned KRAS mutation value, other specific driver mutations have been mentioned as predictive of response and non-response to ICI. EGFR-mutated NSCLC often exhibit compromised responses to anti-PD-(L)1 due to low PD-L1 expression, low mutational burden, and reduced lymphocyte infiltration [73–75]. Similar resistance is observed with ALK rearrangements, MET exon 14, HER2 amplification, RET rearrangement, and less conclusively for BRAF V600E mutations [74,76,77]. STK11/LKB1 mutations also may result in "cold" tumors with poor ICI responses. Lung SCC patients with TP53 and TTN co-mutation had higher TMB levels and better response to ICI [78].

A recent retrospectively work evaluated NSCLC patients treated with ICI in any line and identified 14 gene expression profiles associated with post-ICI survival, namely IDO1, PD-L2, cytotoxicity, cytotoxic cells, IFN downstream, CTLA4, PD-L1, TIGIT, lymphoid, immunoproteasome, exhausted CD8, IFNγ, tumor inflammation signature and APM. In this study, tumor inflammation signature and IFN-γ were the most significant gene expression profiles associated with favorable outcomes [79]. Other gene expression profiles have been proposed, and an additional study found that the TGF-β, dendritic cells, and myeloid signature scores were higher for patients without durable clinical benefit of ICI, whereas the JAK/STAT loss signature scores were higher for patients with durable clinical benefit of ICI [80].

Besides the already mentioned potential role of lactate dehydrogenase levels, other clinical and peripheral blood inflammation-based biomarkers have also been studied in this subject. A prospective observational trial found that high absolute lymphocyte count and absence of liver metastases were associated with durable clinical benefit of ICI in NSCLC [81]. An additional study suggested that high baseline levels of transcription factor T cell factor 1 (TCF1+) CD$_8$$^+$ T cell ratio and peripheral white blood cell count, lymphocyte percentage and cytokeratin 19 fragment after one cycle of ICI treatment

may predict treatment response [82]. Finally, post-treatment eosinophil fraction, systemic immune-inflammation index, modified Glasgow prognostic score, and systemic immune-inflammation index variation were also mentioned as independent predictors of major pathologic response in patients with NSCLC treated with neoadjuvant ICI [81].

**Table 3.** Summary of main identified predictive biomarkers of ICI response in NSCLC.

| Biomarker | Distinctive Characteristic | Prognostic Value |
|---|---|---|
| PD-L1 expression in tumor sample [52,53,59–64] | Positive expression | Select patients who may benefit from ICI treatment |
| | Higher values | Predictor of greater benefit from ICI treatment |
| Soluble or Exosomal PD-L1 [67] | High pre-treatment levels | Unfavorable prognostic factors to patients undergoing ICI treatment |
| Exosomal PD-L1 [67] | Dynamic post-treatment upregulation | Favorable prognosis to patients undergoing ICI treatment |
| TMB [3,70,71] | High-TMB | Select patients who may benefit from ICI treatment |
| mTBI [72] | On-treatment mTBI decreasing levels | Predictor of greater benefit from ICI treatment |
| TILs [68] | High levels of tumor-reactive TILs | Select patients who may benefit from ICI treatment |
| NLR [69] | Dynamic evolution (changed to low) | Favorable prognosis to patients undergoing ICI treatment |
| Specific Mutations [73–78] | EGFR mutations ALK rearrangements MET exon 14 skipping HER2 amplification RET rearrangement KRAS mutations * | Unfavorable prognostic factors to patients undergoing ICI treatment |

ICI, immune checkpoint inhibitor; mTBI, molecular tumor burden index; NLR, neutrophil/lymphocyte ratio; PD-L1, programmed death-ligand 1; TILs, tumor-infiltrating lymphocytes; TMB, tumor mutation burden. * KRAS mutations with STK11/LKB1 and KEAP1 co-mutations.

## 5. Conclusions

The incidence of ICI-induced VLLs probably remains under-reported, which limits this assessment. According to published clinical cases, ICI-induced VLLs in NSCLC appear to relate with a favorable response to ICI. To better understand this potential correlation, more strong data are needed. Two of the three presented clinical cases of ICI-induced VLLs in NSCLC had a favorable response to ICI treatment. The presented case of no response to ICI therapy could potentially be related to KRAS co-mutation status, not available for analysis.

Predictive ICI biomarker identification has become a hot topic among researchers, in line with the precision medicine concept. The understanding of the pathophysiological processes under ICI-induced VLLs may potentially identify predictive biomarkers of ICI response.

**Author Contributions:** Conceptualization, J.Q.C., R.R., M.J.S., S.X.A., P.F. and A.A.; methodology, J.Q.C., S.X.A. and P.F.; validation, J.Q.C., R.R., M.J.S., S.X.A., P.F. and A.A.; formal analysis, J.Q.C., S.X.A. and P.F.; investigation, J.Q.C., S.X.A. and P.F.; resources, J.Q.C., S.X.A. and P.F.; data curation, J.Q.C., S.X.A. and P.F.; writing—original draft preparation, J.Q.C., S.X.A. and P.F.; writing—review and editing, J.Q.C., S.X.A. and A.A.; visualization, J.Q.C., R.R., M.J.S., S.X.A., P.F. and A.A.; supervision, A.A.; project administration, A.A. All authors have read and agreed to the published version of the manuscript.

**Funding:** This research received no external funding.

**Institutional Review Board Statement:** The study was conducted in accordance with the Declaration of Helsinki and approved by the Ethics Committee of UNIDADE LOCAL DE SAÚDE DE SANTO ANTÓNIO (CC 03–RP 26/07/2023).

**Informed Consent Statement:** Written informed consent was obtained from the patients to publish this paper.

**Data Availability Statement:** Not applicable.

**Conflicts of Interest:** The authors declare no conflict of interest.

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
