# Peer review of "Vitiligo-like Lesions as a Predictor of Response to Immunotherapy in Non-Small Cell Lung Cancer: Comprehensive Review and Case Series from a University Center"

_curroncol, doi:10.3390/curroncol31020083_

Round 1

Reviewer 1 Report

Comments and Suggestions for Authors

The paper presents valuable insights into the relationship between immunotherapy-induced Vitiligo-like Lesions (VLL) and treatment outcomes in Non-Small Cell Lung Cancer (NSCLC). I suggest revisions to strengthen the manuscript further.

Major: 

1. Novelty. There are similar topics published recently. Please consider adding these references and try to improve the novelty. 

Case Report: Immune checkpoint inhibitor-related vitiligo-like depigmentation in non-melanoma advanced cancer: A report of three cases and a pooled analysis of individual patient data. PMID: 36713515

Immune checkpoint inhibitor-induced vitiligo in cancer patients: characterization and management. PMID: 36809408

2. Detailed Discussion of Mechanisms: Expanding on the potential mechanisms behind the observed phenomenon of VLL could provide a deeper understanding. 

3. Clinical Practice Implications: A section discussing how these findings might influence future clinical practices in NSCLC treatment would add practical value. 

Comments on the Quality of English Language

Good paper but there are similar topics published. 

Reviewer 2 Report

Comments and Suggestions for Authors

The authors reviewed the association between the development of vitiligo-like lesions (VLL) and the efficacy of immune checkpoint inhibition for NSCLC, presenting examples of experience at their institution. While the cited papers are generally appropriate and well-summarized, there are some points for modification and improvement.

My comments are listed below.

Major comments:

1.        Firstly, the statement in the Abstract, "We present, to our knowledge, the first published case of ICI-induced VLL in NSCLC with no response to ICI treatment, whose reason may be related to the tumor genetic signature. The recognition and understanding of the pathophysiological processes underlying ICI-induced VLL may represent a promising opportunity to identify a predictive marker of tumor response to ICI, with an impact on treatment selection and patient management. It also may contribute to the recognition of new patterns of molecular expression that could lead to improvements in therapeutic development (lines 32-38)," is deemed inappropriate and should be considered for deletion and revision. It appears to be based on the experience of Case 3, but the lack of response to ICI treatment in this patient may be attributed not only to KRAS but also to factors such as 0% expression of PD-L1 and unknown background factors. In particular, the phrase "We present, to our knowledge, the first published case of ICI-induced VLL in NSCLC with no response to ICI treatment, whose reason may be related to the tumor genetic signature" is considered highly inappropriate. Asserting that the authors are presenting the first case despite uncertain information is unacceptable. Similar content is found in "4.4 Reflection on Reported Experience," where the statement "To our knowledge, we present the first published case of ICI-induced VLL in NSCLC without clinical or imagological response to treatment (case report 3). The reason behind this outcome remains unknown" appears. Due to the same reasons, please reconsider all statements in Section 4.4.

2.        The authors have not included many figures and tables in their Review. If possible, it would be better to use figures and tables in the sections that can be explained with figures and tables, so that readers can more easily understand what the authors want to show in the Review. Please reconsider.

Round 2

Reviewer 1 Report

Comments and Suggestions for Authors

Thank you

Comments on the Quality of English Language

minor.

Reviewer 2 Report

Comments and Suggestions for Authors

I judged that the authors were responding appropriately to my comments.

I appreciate the authors' appropriate revisions.